# Effects of Monotherapy with Clopidogrel vs. Aspirin on Vascular Function and Hemostatic Measurements in Patients with Coronary Artery Disease: The Prospective, Crossover I-LOVE-MONO Trial

**DOI:** 10.3390/jcm10122720

**Published:** 2021-06-20

**Authors:** Hyun-Woong Park, Min-Gyu Kang, Jong-Hwa Ahn, Jae-Seok Bae, Udaya S. Tantry, Paul A. Gurbel, Young-Hoon Jeong

**Affiliations:** 1Department of Internal Medicine, Gyeongsang National University School of Medicine and Gyeongsang National University Hospital, Jinju 52727, Korea; chunjium@hanmail.net (H.-W.P.); med2floyd@naver.com (M.-G.K.); 2Department of Internal Medicine, Gyeongsang National University School of Medicine and Cardiovascular Center, Gyeongsang National University Changwon Hospital, Changwon 51472, Korea; jonghwaahn@naver.com (J.-H.A.); baefach@naver.com (J.-S.B.); 3Sinai Center for Thrombosis Research and Drug Development, Sinai Hospital of Baltimore, Baltimore, MD 21215, USA; ukstantry@gmail.com (U.S.T.); Pgurbel@lifebridgehealth.org (P.A.G.); 4Institute of the Health Sciences, Gyeongsang National University, Jinju 52727, Korea

**Keywords:** aspirin, clopidogrel, platelet, endothelium, coagulation

## Abstract

Objectives: To evaluate the effect of clopidogrel vs. aspirin monotherapy on vascular function and hemostatic measurement. Background: Monotherapy with P2Y_12_ receptor inhibitor vs. aspirin can be a useful alterative to optimize clinical efficacy and safety in high-risk patients with coronary artery disease (CAD). Methods: We performed a randomized, open-label, two-period crossover study in stented patients receiving at least 6-month of dual antiplatelet therapy (DAPT). Thirty CAD patients with moderate-to-high ischemic risk were randomly assigned to receive either 75 mg of clopidogrel or 100 mg of aspirin daily for 4 weeks, and were crossed over to the other strategy for 4 weeks. Vascular function was evaluated with reactive hyperemia-peripheral arterial tonometry (RH-PAT) and brachial-ankle pulse wave velocity (baPWV). Hemostatic profiles were measured with VerifyNow and thromboelastography (TEG). The primary endpoint was the reactive hyperemia index (RHI) during clopidogrel or aspirin monotherapy. Results: Clopidogrel vs. aspirin monotherapy was associated with better endothelial function (RHI: 2.11 ± 0.77% vs. 1.87 ± 0.72%, *p* = 0.045), lower platelet reactivity (130 ± 64 vs. 214 ± 50 P2Y12 reaction unit [PRU], *p* < 0.001) and prolonged reaction time (TEG R: 5.5 ± 1.2 vs. 5.1 ± 1.1 min, *p* = 0.037). In multivariate analysis, normal endothelial function (RHI ≥ 2.1) was significantly associated with clot kinetics (TEG angle ≤ 68 degree) and ‘PRU ≤ 132’. ‘PRU ≤ 132’ was achieved in 46.2% vs. 3.8% during clopidogrel administration vs. aspirin monotherapy (odds ratio 21.4, 95% confidence interval 2.7 to 170.1, *p* < 0.001). Conclusions: In CAD patients, clopidogrel vs. aspirin monotherapy was associated with better endothelial function, greater platelet inhibition and lower coagulation activity, suggesting pleiotropic effects of clopidogrel on endothelial function and hemostatic profiles.

## 1. Introduction

Aspirin monotherapy following 6–12 month dual antiplatelet therapy (DAPT) has been a major secondary prevention strategy in patients with coronary artery disease (CAD) treated with percutaneous coronary intervention (PCI) [1]. However, long-term aspirin monotherapy in low-risk patients has shown limited effect in reducing ischemic events but an increased risk of gastrointestinal bleeding [2]. A recent meta-analysis demonstrated that P2Y_12_ inhibitor monotherapy is associated with a reduced risk of myocardial infarction (MI) and a comparable rate of stroke in patients with established atherosclerotic cardiovascular disease (ASCVD) compared to aspirin monotherapy [3].

Long-term DAPT can be associated with an increased bleeding risk [1,4]. Therefore, monotherapy with a P2Y_12_ inhibitor following standard duration of DAPT in PCI-treated patients may be a credible alternative strategy [5]. Although this strategy has demonstrated better clinical outcomes with reduced bleeding risk compared to DAPT [6,7], its clinical benefit was proven for a limited time period (mostly within 6–12 months). A single-center experience suggested that monotherapy with clopidogrel vs. aspirin after 12 months of DAPT was associated with a reduced risk of ischemic events in CAD patients treated with drug-eluting stent (DES) [8]. Large-scale clinical trials have been under investigation to evaluate the efficacy and safety of clopidogrel monotherapy beyond 6–18 months in DES-treated patients [9,10].

Endothelial dysfunction is a cardiovascular (CV) risk factor and is strongly associated with future CV events [11]. P2Y_12_ inhibitors exhibit pleiotropic effects on vascular function and hemostatic profiles through multiple pathways [12]. Previous studies suggested that adjunctive use of P2Y_12_ inhibitor to aspirin was associated with better endothelial function and low level of inflammation in CAD patients [13,14], but few studies directly compared the effect of clopidogrel vs. aspirin monotherapy on vascular and hemostatic measurements in CAD patients.

Therefore, we sought to assess the impact of clopidogrel- vs. aspirin-monotherapy on endothelial function and hemostatic assay in DES-treated CAD patient with moderate-to-high ischemic risk after standard DAPT duration.

## 2. Materials and Methods

The I-LOVE-MONO (Impact of cLOpidogrel VErsus aspirin MONOtherapy on endothelial function) trial was a prospective, randomized, open-label, two-center, two-way superiority crossover study conducted in South Korea between August 2018 and April 2019. The study protocol and the informed consent form were approved by the institutional review boards of the hospitals (GNUH-2018-01-013, CGNUH-2017-18-005). Informed written consent was obtained from all patients, and the study was performed in accordance with the Good Clinical Practice Guidelines and the principles of the Declaration of Helsinki [15].

### 2.1. Study Population

The enrolled patients underwent PCI with angioplasty and DES implantation, and were on 100 mg of aspirin and 75 mg of clopidogrel once a day for at least 6 months. The exclusion criteria were: (1) a history of active bleeding and bleeding diatheses; (2) concomitant oral anticoagulation therapy; (3) hemodynamic instability; (4) left ventricular ejection fraction < 40%; (5) leukocyte count < 3000/mm^3^ and/or platelet count < 100,000/mm^3^; (6) aspartate aminotransferase or alanine aminotransferase level > 3 times the respective upper normal limits; (7) serum creatinine level > 3.5 mg/dL; (8) stroke within 3 months; and (9) non-cardiac disease with a life expectancy < 1 year.

### 2.2. Study Design

The study included a screening phase and two 4-week treatment periods (Figure 1). Eligible patients were randomly assigned to receive antiplatelet monotherapy with clopidogrel or aspirin for 4 weeks (±3 days): 75 mg of clopidogrel (Pidogul^®^; Hanmi Pharmaceutical Co., Ltd., Seoul, South Korea) once daily (CLPD) or 100 mg of aspirin (Aspirin Protect^®^; Bayer Korea Co., Ltd., Seoul, South Korea) once daily (ASP) and were then crossed over to the other regimen for 4 more weeks. No other changes in medications were allowed during the study period. At the end of each treatment period, the patients’ adherence and adverse event occurrences were recorded by the attending physician based on face-to-face interviews, pill counting and a dedicated questionnaire.

### 2.3. Hemostatic Measurements

Immediately before randomization (baseline) and at the end of each study period, blood samples were obtained between 2 and 6 h after the last study-drug administration from the antecubital vein after discarding first 2 mL of free-flowing blood and measurements were conducted within 2 h of blood sampling.

#### 2.3.1. VerifyNow Assay

The VerifyNow P2Y12 assay (Accriva, San Diego, CA, USA) is a whole-blood, point-of-care, turbidimetric-based optical detection assay designed to measure platelet aggregation in response to an adenosine diphosphate (ADP) [16]. Blood samples were collected in 3.2% citrate Vacuette tubes (Greiner Bio-One Vacuettew North America, Inc., Monroe, NC, USA). The cartridge consists of two channels. One channel contains fibrinogen-coated polystyrene beads, 20 μM ADP and 22 nM prostaglandin E1, of which the optical signal is reported as P2Y12 Reaction Unit (PRU). Another second channel contains fibrinogen-coated polystyrene beads, 3.4 mM iso-thrombin receptor-activating peptide (protease-activated receptor-1 [PAR-1] agonist) and PAR-4-activating peptide, of which the optical signal is reported as BASE (estimated maximal platelet function independent of P2Y_12_ receptor blockade).

#### 2.3.2. Thromboelastography (TEG)

The assay was performed using the TEG^®^ 5000 Analyzer System (Hemonetics Corp, BrainTree, MS, USA) with automated analytical software, provides measurements of the viscoelastic properties of the clot generation. Blood samples were drawn into Vacutainer tubes containing 3.2% trisodium citrate (Becton-Dickinson, Franklin Lakes, NJ, USA). In brief, 500 μL citrate blood was mixed by inversion with kaolin and 340 μL of activated blood was transferred to a reaction cup, to which 20 μL of 200 mM of calcium chloride was added. The hemostatic measurements were performed as described earlier [17].

All TEG parameters were recorded from standard tracings (Table 1). The reaction time (R, min), a representative of the initiation phase of enzymatic clotting, is the time from the start of the sample run to the point of the first significant clot formation, corresponding to an amplitude of 2 mm reading on the TEG tracing. K is a measure of the time to reach 20 mm clot strength from R. The angle (α) is reflective of the fibrinogen activity and is the degrees of the angle formed by the tangent line to TEG tracing measure at R. Kaolin-induced maximum amplitude (MA_thrombin_, mm) represents the maximum platelet–fibrin clot strength (P-FCS) and is influenced by fibrinogen levels, platelet counts and platelet function. LY30 indicates the percentage of the clot that has lysed in 30 min after the occurrence of MA, and indicates the level of fibrinolytic activity [17].

### 2.4. Vascular Function

#### 2.4.1. Reactive Hyperemia-Peripheral Arterial Tonometry (RH-PAT)

Vascular function was assessed using the EndoPAT^®^ system (Itamar Medical, Caesarea, Israel, software version 3.0.4). The assay provides digital flow-mediated dilation (endothelial function) during reactive hyperemia using measurements from both arms (occluded and control sides) [18]. In brief, the device measures changes of blood volume in the distal finger that accompanies pulse waves. A blood pressure cuff is placed on one upper arm, while the contralateral arm served as a control value. PAT probes are placed on one finger of each hand. After a 5-min resting period, the cuff is inflated to 60 mmHg greater than the systolic pressure or 200 mmHg for 5 min and then deflated to induce reactive hyperemia. The RH-PAT data are digitally analyzed online by a computer in an operator-independent manner. The RH-PAT value is calculated as the ratio of the average amplitude of PAT signal over 1 min starting 1.5 min after cuff deflation (control arm, A; occluded arm, C) divided by the average amplitude of PAT signal of a 2.5-min time period before cuff inflation (baseline) (control arm, B; occluded arm, D).

EndoPAT provides an index of endothelial function in two forms: RHI and LnRHI. RHI is the post-to-preocclusion PAT signal ratio in the occluded side, normalized to the control side and further corrected for baseline vascular tone. LnRHI is a similar index after natural log transformation with a matched cutoff, which provides a better double-sided distribution (closer to normal distribution) than RHI. The criteria of normal endothelial function are defined as ‘RHI ≥ 2.10’ [18].

#### 2.4.2. Arterial Stiffness Indices

To evaluate the arterial stiffness (media function), brachial-ankle pulse wave velocity (baPWV) and augmentation index (AI) were used [19]. PWV is the distance traveled (ΔD) by the wave divided by the time (ΔT) for the wave to travel this distance: PWV = ΔD/ΔT. BaPWV was measured using VP-1000 (Colin Co., Ld., Komaki, Japan), which is an automated and noninvasive device. The four blood pressure cuffs are applied at both ankles and both upper arms, and the pressure wave is acquired simultaneously. The baPWVs are estimated automatically from waveforms at the upper arm and ankle and a patient’s height.

Augmentation index (AI) is derived from pulse waveform analysis. AI was measured using HEM-9000AI (Omron Healthcare Co., Ltd., Kyoto, Japan), which can measure central blood pressure (CBP) by a non-invasive method. The pulse waveform is a composite of forward wave and reflected wave. In conditions of central arterial stiffness, the reflection wave is derived faster, resulting in an augmentation (AG) of CBP. The pulse pressure (PP) is calculated by subtracting the diastolic BP (DBP) from the systolic BP (SBP). AG is the difference between the first and second systolic peak in the central pulse waveform. AI can be calculated by the followed equation: AG/PP × 100%. Because of the AI dependency on heart rate (HR), AI is adjusted to 75 beats (AI@75). The AI@75 is obtained automatically using the following equation: AI + [0.48 × (75 − HR)].

### 2.5. Endpoints

The primary endpoint was RHI or LnRHI (EndoPAT^®^) during monotherapy with clopidogrel vs. aspirin. Secondary endpoints were: (1) platelet reactivity (PRU and BASE); (2) TEG^®^ indices; and (3) arterial stiffness indices. In addition, we evaluated the relationships between vascular function and hemostatic measurements.

### 2.6. Statistical Analysis

The sample size calculation was derived from an earlier experience [13], where adjunctive clopidogrel in addition to aspirin relatively increased the RHI value by 19.7% in CAD patients compared to the control group. At least 26 patients were needed to detect a relatively difference of 19.7% with a power of 90%, a two-sided α error = 0.05, and a standard deviation (SD) of 0.30. Considering a 15% drop-out rate, we enrolled 30 patients.

Normal distribution of measurements was confirmed by a Kolmogorov–Smirnov test. Continuous variables were presented as mean ± SD or as median (interquartile range (IQR)), as appropriate, while categorical variables were reported as frequencies and percentages. Student’s unpaired *t* test for parametric continuous variables and the Wilcoxon signed rank test for non-parametric continuous variables were used. Comparisons between categorical variables were performed using the Pearson Chi-square test or Fisher exact test, as appropriate. Comparison of the antiplatelet effect between the regimens was performed on the per-protocol analysis, defined as all patients who were randomized to receive treatment and finished the study schedule without protocol violation.

Receiver-operating characteristic (ROC) curve analysis was performed to determine optimal cutoffs of continuous variables, which were then changed into the dichotomous covariates. All vascular function and hemostatic measurements were evaluated in a univariate analysis for predicting the determinants of normal endothelial function (RHI ≥ 2.1). Variables with *p* value < 0.1 in the univariate analysis were then entered into the multivariate logistic regression analysis using backward stepwise elimination to provide an odds ratio (OR) and 95% confidence interval (CI). Statistical analyses were performed using SPSSv24.0 software (SPSS Inc., Chicago, IL, USA). A *p* < 0.05 was considered statistically significant.

## 3. Results

### 3.1. Study Population

Thirty Korean patients treated with DES were initially enrolled. Twenty-six subjects completed both periods of the study and showed complete adherence, whereas four subjects were unable to complete the study protocol because of withdrawal (*n* = 2), poor adherence (*n* = 1) and a technical problem (*n* = 1) during the first period (Figure 1). There were no serious adverse events during the protocol period.

The vast majority of patients were men (*n* = 24; 92.3%) and were treated with ≥ 12 months of DAPT (*n* = 21; 80.8%) (9 months, *n* = 2; 10 months, *n* = 1; 11 months, *n* = 2) (Table 2). All patients had at least one covariate in ‘moderate-to-high risk of ischemic events’ (multivessel disease, *n* = 15; history of myocardial infarction, *n* = 15; diabetes mellitus, *n* = 8; chronic kidney disease, *n* = 1; peripheral artery disease, *n* = 3; and heart failure, *n* = 1) [4].

### 3.2. Primary Endpoint

RHI values during clopidogrel monotherapy were significantly higher compared to aspirin monotherapy (2.11 ± 0.77 vs. 1.87 ± 0.72; difference, 0.24; △95% CI, 0.01 to 0.47; *p* = 0.045) (Figure 2A, Table 3). Likewise, there was a significant difference in LnRHI levels between monotherapy with clopidogrel vs. aspirin (0.68 ± 0.37 vs. 0.55 ± 0.35; difference, 0.13; △95% CI, 0.03 to 0.23; *p* = 0.015) (Table 3). RHI and LnRHI values during clopidogrel monotherapy were similar to those during DAPT (all *p* values ≤ 0.479).

### 3.3. Secondary Endpoint

Clopidogrel monotherapy showed a lower level of PRUs compared to aspirin monotherapy (130 ± 64 vs. 214 ± 50 PRU: difference, −84 PRU; △95% CI, −110 to −58 PRU; *p* < 0.001) (Figure 2B, Table 3). There were no differences in the TEG^®^ indices except the R value (clopidogrel vs. aspirin: 5.5 ± 1.2 vs. 5.1 ± 1.1 min; difference, 0.5 min; △95% CI, 0.1 to 0.9 min; *p* = 0.037) between the two treatments. VerifyNow and TEG^®^ indices were similar when treated with clopidogrel monotherapy and DAPT (0.102 ≤ *p* values ≤ 0.748).

There were no significant differences in terms of arterial stiffness indices according to monotherapy regimen (all *p* values ≥ 0.282) (Table 3). In addition, DAPT and clopidogrel monotherapy strategies showed the similar values.

### 3.4. Determinants for Normal Endothelial Function (RHI ≥ 2.1)

After pooling data from individual antiplatelet treatment together (78 sets), we compared arterial stiffness and hemostatic measurements according to the status of endothelial function (Appendix A). Clopidogrel administration (DAPT or clopidogrel monotherapy) was associated with an increased chance of normal endothelial function (RHI ≥ 2.1) compared to aspirin monotherapy (44.2% vs. 19.2%: OR, 3.331; 95% CI, 1.089 to 10.192; *p* = 0.030).

The correlation between PRU and RHI was weak (r = −0.119) (Figure 3A). In the ROC curve analysis to assess the PRU value corresponding to normal endothelial function (RHI ≥ 2.1), ‘PRU ≤ 132′ was identified as the matched cutoff (area under curve, 0.626; 95% CI, 0.490 to 0.762; *p* = 0.065), with a sensitivity of 78.0% and a specificity of 50.0% (Figure 3B).

In a multivariate analysis, ‘PRU ≤ 132′ (OR, 4.028; 95% CI, 1.087 to 14.925; *p* = 0.037) and ‘TEG angle ≤ 68 degree’ (OR, 7.420; 95% CI, 1.445 to 38.099; *p* = 0.016) were significantly associated with normal endothelial function (Table 4). Clopidogrel administration (DAPT or clopidogrel monotherapy) showed a higher chance of ‘PRU ≤ 132’ compared to aspirin monotherapy (46.2% vs. 3.8%: OR, 21.429; 95% CI, 2.699 to 170.124; *p* < 0.001).

## 4. Discussion

To the best of our knowledge, the I-LOVE-MONO trial was the first investigation to compare vascular function (endothelial function and arterial stiffness indices) and hemostatic measurements (platelet function and global hemostasis assay) during monotherapy with clopidogrel vs. aspirin in CAD patients with moderate-to-high ischemic risk. The main findings of the study are as follows: (1) clopidogrel monotherapy showed a better endothelial function (RHI, LnRHI) compared to aspirin monotherapy; (2) potent antiplatelet effect (PRU ≤ 132) increased the chance of normal endothelial function (RHI ≥ 2.1) by ~4-fold; and (3) monotherapy with clopidogrel vs. aspirin was associate with a prolonged coagulation time (indicated by TEG R).

### 4.1. Monotherapy with P2Y_12_ Inhibitor vs. Aspirin in ASCVD Patients

Optimal long-term medical therapy and control of risk factors can improve prognosis in chronic coronary syndrome (CCS) with a high-risk profile. A recent clinical guideline recommends that vascular-dose rivaroxaban or a P2Y_12_ inhibitor in addition to aspirin should be considered in CAD patients who are at moderate-to-high risk of ischemic events and low risk of major bleeding [4]. In the randomized PEGASUS-TIMI 54 (Prevention of Cardiovascular Events in Patients With Prior Heart Attack Using Ticagrelor Compared to Placebo on a Background of Aspirin–Thrombolysis in Myocardial Infarction 54) trial (*n* = 21,162), the use of low-dose ticagrelor (60 mg twice a day) plus aspirin after 1–3 years from MI reduced the risk of ischemic events, at the price of limited increase on serious bleeding complications [20]. In addition, a recent Western real-world observational study provided that low-dose ticagrelor after 12 months from MI showed to be effective and safe, with no major bleeding occurring at follow-up [21].

However, this dual-pathway inhibition may increase the risk of bleeding in patients with high bleeding risk (HBR). Therefore, we need to emphasize the de-escalation strategy (e.g., aspirin-free strategy with P2Y_12_ inhibitor monotherapy) to maximize long-term net clinical benefit in these subjects [5]. In East Asian patients, there is an emerging concern regarding the increased risk of serious bleeding during antithrombotic treatment, which emphasizes the mandatory application of the de-escalation strategy during the chronic phase [22]. For example, the PEGASUS subgroup analysis from Asian patients showed no difference in the incidence of ischemic events (7.11% vs. 6.86%) but a considerable increase in the number of bleeding episodes (3.74% vs. 1.44%) at 3 years after treatment with 60 mg of ticagrelor vs. placebo in addition to aspirin [23].

In a meta-analysis of secondary prevention trials including patients with ASCVD of cerebrovascular, coronary or peripheral system (nine trials, *n* = 42,108), monotherapy with a P2Y_12_ inhibitor showed a lower risk of MI (OR, 0.81; 95% CI, 0.66 to 0.99) and a similar risk of major bleeding (OR, 0.90; 95% CI, 0.74 to 1.10) compared to monotherapy with aspirin, and these findings were consistent regardless of the type of P2Y_12_ inhibitor used [3]. For example, the CAPRIE (clopidogrel versus aspirin in patients at risk of ischemic events) study including patients with established disease of ischemic stroke, MI or symptomatic peripheral artery disease (PAD) (*n* = 19,185) showed that clopidogrel monotherapy reduced the risk of ischemic events and vascular deaths compared to aspirin monotherapy (5.32% vs. 5.83% during a mean follow-up of 1.91 years; relative risk reduction 8.7%; *p* = 0.043). Monotherapy with clopidogrel vs. aspirin showed less frequent risks of bleeding and gastrointestinal problems [24].

Among East Asian patients presented with ACS, long-term use (over 1 year) of standard-dose potent P2Y_12_ inhibitor (e.g., 10 mg of prasugrel and 60 mg of ticagrelor) in addition to aspirin has shown to increase the risk of major bleeding without reducing the risk of ischemic events compared to clopidogrel treatment [25]. It remains uncertain which P2Y_12_ inhibitor monotherapy following DAPT would be the best option for a long-term strategy for secondary prevention in East Asian patients with a moderate-to-high risk profile. A single-center experience including DES-treated patients receiving 12-month DAPT showed that clopidogrel monotherapy (*n* = 771) was associated with a reduction in risk for a composite of cardiac death, MI or stroke (2.6% vs. 3.8%; HR, 0.54; 95% CI, 0.32 to 0.92; *p* = 0.02) compared to aspirin monotherapy. However, TIMI (thrombolysis in myocardial infarction) major bleeding was similar between both groups (1.3% vs. 0.9%; HR, 1.03; 95% CI, 0.46 to 2.32; *p* = 0.95) [8]. Recently, the HOST-EXAM (Harmonizing Optimal Strategy for Treatment of coronary artery stenosis-Extended Antiplatelet Monotherapy) trial compared clinical efficacy and safety of clopidogrel vs. aspirin monotherapy in DES-treated patients (*n* = 5530) who maintained DAPT without clinical events for 6–18 months. At the two-year follow-up, clopidogrel monotherapy was associated with both reduction in thrombotic events (3.7% vs. 5.5%; HR, 0.68; 95% CI, 0.52 to 0.87; *p* = 0.003) and any bleeding episodes (≥BARC type 2: 2.3% vs. 3.3%; HR, 0.70; 95% CI, 0.51 to 0.98; *p* = 0.036) compared to aspirin monotherapy [9].

### 4.2. Effect of P2Y_12_ Receptor Inhibitor on Endothelial Function

Vascular failure has been defined as the integration of vascular endothelial dysfunction, smooth muscle cells dysfunction and metabolic dysfunction. Endothelial function can be measured by flow-mediated vasodilation (FMD) in the brachial artery and RH-PAT in the fingertip. Arterial stiffness, which is reflective of medial layer function, can be assessed by PWV and cardio-ankle vascular index [18]. The endothelium plays a pivotal role in maintaining vascular homeostasis by regulating factors that promote or inhibit multiple processes, such as vasoconstriction, vasodilation, cell growth, thrombosis, inflammation and oxidation. Endothelial dysfunction is mainly associated with the attenuated bioavailability of vasodilator factors (e.g., nitric oxide [NO]), and initiates atherogenesis and atherothrombotic complications [18]. The latter can predict future CV events and all-cause mortality [11,18]. Endothelial dysfunction should be an important potential target of preventive CV strategies.

There is accumulating evidence that P2Y_12_ inhibitors exert extra-platelet effects. P2Y_12_ receptors can also be found in a wide variety of tissues, including certain subregions of the brain, vascular smooth muscle cells, leukocytes, macrophages, microglia and dendritic cells, which may indicate the potential of extra-platelet effect by P2Y_12_ receptor blockers [12]. Furthermore, these pleiotropic effects of P2Y_12_ receptor inhibitors may also derive from different mechanisms other than the inhibition of ADP-induced platelet aggregation. In a heart failure rat model, increasing sensitivity of P2Y_12_ signaling leads to impaired adenylyl cyclase-mediated signaling and NO bioavailability, which were associated with endothelial dysfunction and enhanced platelet reactivity. Chronic P2Y_12_ blockade with P2Y_12_ inhibitor improved adenylyl cyclase-mediated signaling, resulting in improved endothelial function and NO bioavailability [26]. Furthermore, clopidogrel-derived S-nitrosothiol (SNO: more stable than NO) was formed directly from the base drug without the need for prior in vivo metabolism [27]. In CAD patients on long-term aspirin treatment, 5-week adjunctive use of clopidogrel improved endothelial NO bioavailability (improvement of acetylcholine-induced vasodilatation and L-NMMA responses) and decreased biomarkers of oxidant stress and inflammation compared to placebo [28].

Endothelial cells have important role in NO-mediated regulation of platelet activation [29]. In addition, platelet-derived NO inhibits platelet aggregation by increasing cGMP. Fujisue et al. demonstrated that RHI was negatively correlated with PRU (r = −0.32, *p* = 0.001), and this indicator was an independent determinant of high platelet reactivity (PRU ≥ 230) (OR, 0.55; 95% CI, 0.39 to 0.78; *p* = 0.001) in CAD patients during DAPT [30]. However, another study suggested no significant correlation between endothelial function and platelet function [13]. In the current study, PRU showed a weak association with RHI (r = −0.119), but low platelet reactivity (PRU ≤ 132) was significantly associated with the normal endothelial function (RHI ≥ 2.1) (OR, 4.03; 95% CI, 1.09 to 14.93; *p* = 0.037), which again suggested the close interaction between endothelial function and platelet reactivity.

There are conflicting results regarding the relationship between the type of P2Y_12_ inhibitor and endothelial function. In diabetic patients presented with non-ST-segment elevation ACS [31], ticagrelor showed better improvement of endothelial function (indicated by FMD) compared to prasugrel, which has been explained by an increased adenosine level. However, other studies reported that ticagrelor did not improve endothelial function compared to prasugrel and clopidogrel [32]; rather, endothelial function was improved during prasugrel treatment [33].

### 4.3. Relationship between TEG Measurements and Platelet/Endothelial Function

During the initial phase of thrombus generation, platelets express phosphatidylserine, P-selectin and CD-40L, which facilitate the formation of platelet-monocyte aggregates. Both the phosphatidylserine and platelet-monocyte aggregates provide a friendly environment for the expression of tissue factor and enhance the rate of thrombin generation [34]. In addition, P2Y receptors, especially P2Y_12_ receptors, play important roles in the procoagulant activity of platelets [35]. Clopidogrel loading before PCI was associated with a prolongation in R value (4.4 ± 1.4 vs. 5.4 ± 1.7 min at pre- vs. post-loading; *p* < 0.001), which directly correlated with the change in platelet aggregation (r = 0.65, *p* < 0.0001) [34]. Another study evaluated the impact of adjunctive antiplatelet regimen on platelet reactivity and biomarkers in patients with atrial fibrillation on aspirin. Adjunctive use of clopidogrel or cilostazol reduced levels of the von Willebrand factor and fibrinogen, of which changes were mainly associated with the change in ADP-mediated platelet reactivity [36]. The present study again demonstrated that the time to initial platelet-fibrin clot formation (R time), a representative of the initiation phase of enzymatic clotting, was prolonged during clopidogrel vs. aspirin monotherapy, which suggests a close relationship between the inhibition of platelet P2Y_12_ receptors and procoagulant activity.

In TEG measurement, the alpha angle (rate of clot development) reflects the speed of fibrin accumulation and polymerization, which is influenced by fibrinogen levels and use of anticoagulant [37]. Fibrinogen activity is mechanistically linked with the inflammation state. In addition, fibrinogen alters the vascular reactivity and impairs endothelial cell layer integrity by binding to its endothelial cell membrane receptors [38]. Therefore, the level of alpha angle may be related to the risk of endothelial dysfunction. This study revealed that a lower alpha angle (e.g., ≤ 68 degrees) was an independent determinant for normal endothelial function.

### 4.4. Study Limitations

First, the present study was conducted with a small number of patients with a short-term treatment. Therefore, its influence on clinical efficacy and safety are inconclusive. Second, our study included only East Asian patients. Because East Asian patients have a lower response to clopidogrel treatment compared to Western patients, the ability to generalize the results from the current study to a Western population may be limited. Finally, monotherapy with a potent P2Y_12_ inhibitor was not tested in the present study.

## 5. Conclusions

In CAD patients with moderate-to-high ischemic risk, monotherapy with clopidogrel vs. aspirin was associated with better endothelial function, greater platelet inhibition and lower coagulation activity, which may suggest the various pleiotropic effects of clopidogrel on endothelial function and hemostatic profiles. These pleiotropic effects of clopidogrel may inhibit atherothrombotic progression and improve future CV outcomes.

## Figures and Tables

**Figure 1 jcm-10-02720-f001:**
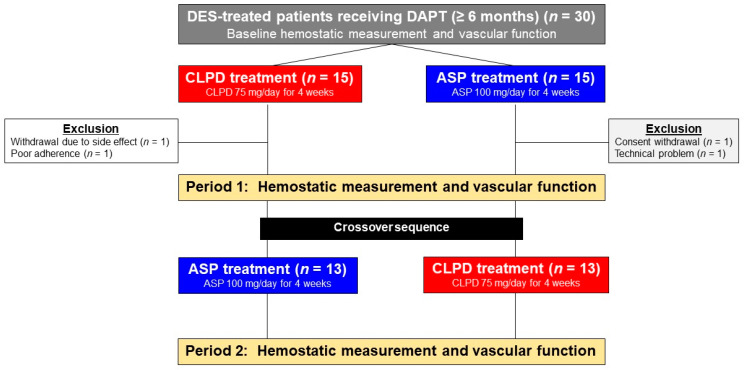
Flow diagram of the study. ASP, aspirin; CLPD, clopidogrel; DAPT, dual antiplatelet therapy; DES, drug-eluting stent.

**Figure 2 jcm-10-02720-f002:**
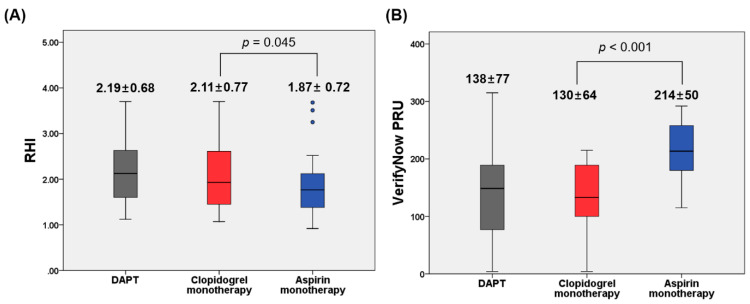
EndoPAT (**A**) and VerifyNow assay (**B**) according to antiplatelet regimen. The central box represents the values between the lower and upper quartiles, and the middle line is the median. The vertical line extends from the minimum to the maximum value, excluding outside values, which are displayed as separate points. PRU, P2Y12 Reaction Units; RHI, reactive hyperemia index.

**Figure 3 jcm-10-02720-f003:**
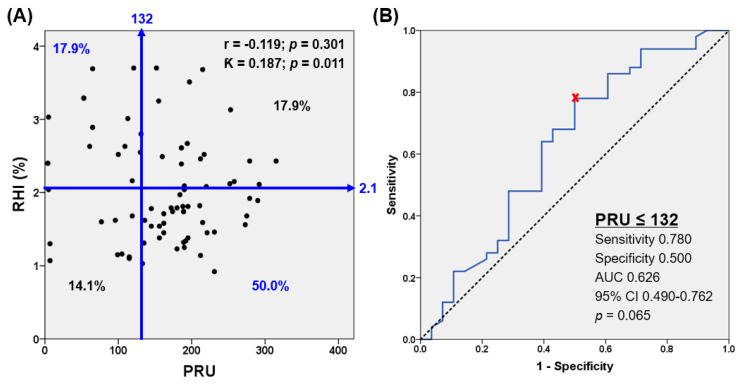
Relation between platelet reactivity and RHI (**A**) and ROC curve analysis for platelet reactivity corresponding to normal endothelial function (RHI ≥ 2.1) (**B**). AUC, area under curve; CI, confidence interval; ROC, receiver operating characteristic.

**Table 1 jcm-10-02720-t001:** Thromboelastographic indices.

R, minutes	Time from the initiation of the until the point where the clot begins to form
K, minutes	Interval from the split point of the test to the point where the fibrin cross-linking provides enough clot resistance to produce a 20-mm amplitude
Angle, degree	Angle formed by the slope of a tangent line traced from the R time to the K time reflects the rate at which the clot forms
MA, mm	Maximum amplitude of the clot dynamics, reflecting ‘platelet-fibrin clot strength’
LY30, %	Percentage of the clot that has lysed 30 min after the time of MA

**Table 2 jcm-10-02720-t002:** Baseline characteristics.

Variables	(*n* = 26)
Duration of DAPT, months	12 (11, 17)
Age, year old	60.4 ± 7.6
Male, *n* (%)	24 (92.3)
BMI, kg/m^2^	25.4 ± 2.5
Index disease entity	
Stable angina	5 (19.2)
Unstable angina	6 (23.1)
Non-ST-segment elevation MI	10 (38.5)
ST-segment elevation MI	5 (19.2)
Risk factors or past history, *n* (%)	
Diabetes mellitus	8 (30.8)
Hypertension	14 (53.8)
Dyslipidemia	15 (57.7)
Current smoking	7 (26.9)
Chronic kidney disease	1 (3.8)
Peripheral artery disease	3 (11.5)
Previous stroke	2 (7.7)
Concomitant medication, *n* (%)	
Aspirin	26 (100)
Clopidogrel	26 (100)
Statin	26 (100)
Beta blocker	20 (76.9)
Angiotensin blockade	22 (84.6)
Calcium channel blocker	6 (23.1)
Proton pump inhibitor	21 (80.8)
Laboratory data	
WBC count, ×10^3^/mm^3^	6.6 ± 1.9
Hemoglobin, g/dL	14.6 ± 1.2
Platelet count, ×10^3^/mm^3^	229 ± 34
Hb_A1C_, %	6.0 ± 0.6
GFR, ml/min/1.73 m^2^ (MDRD)	90 ± 18
Total cholesterol, mg/dL	135 ± 38
LV ejection fraction, %	60 ± 6
Procedural data	
Multivessel disease, *n* (%)	15 (57.7)
Multivessel stenting, *n* (%)	9 (34.6)
Drug-eluting stent implantation, *n* (%)	26 (100)
Target lesion, *n* (%)	
Left anterior descending	17 (65.4)
Left circumflex	2 (7.7)
Right coronary	7 (26.9)
Stent number	1.7 ± 0.7

DAPT, dual antiplatelet therapy; GFR, glomerular filtration rate; HbA1c, hemoglobin A1c; LV, left ventricular; MDRD, Modification of Diet in Renal Disease; MI, myocardial infarction; WBC, white blood cell.

**Table 3 jcm-10-02720-t003:** Vascular function and hemostatic measurement.

	DAPT (CLPD+ASP)	Monotherapy (CLPD)	Monotherapy (ASP)	*p* (DAPT vs. CLPD)	*p* (DAPT vs. ASP)	*p* (CLPD vs. ASP)
**EndoPAT**						
RHI, %	2.19 ± 0.68	2.11 ± 0.77	1.87 ± 0.72	0.479	0.023	0.045
LnRHI	0.74 ± 0.32	0.68 ± 0.37	0.55 ± 0.35	0.331	0.002	0.015
**Arterial stiffness indices**						
Brachial SBP, mmHg	132.9 ± 17.3	130.0 ± 18.1	125.5 ± 19.6	0.559	0.152	0.386
Brachial DBP, mmHg	76.2 ± 10.3	74.2 ± 10.1	73.4 ± 11.0	0.491	0.346	0.773
Brachial PP, mmHg	56.0 ± 10.7	55.9 ± 12.0	52.1 ± 12.7	0.961	0.239	0.282
Central SBP, mmHg	138.7 ± 19.7	134.7 ± 20.7	130.4 ± 21.6	0.474	0.152	0.466
Central DBP, mmHg	76.2 ± 10.3	74.2 ± 10.1	73.4 ± 11.0	0.491	0.346	0.773
Central PP, mmHg	62.6 ± 14.0	60.5 ± 14.1	57.0 ± 13.9	0.597	0.159	0.377
AI, %	83.5 ± 12.2	81.6 ± 12.4	81.6 ± 11.6	0.583	0.570	1.000
AI@75, %	77.0 ± 11.7	75.4 ± 11.6	75.8 ± 11.1	0.602	0.690	0.893
PWV (mean), m/sec	15.9 ± 2.8	15.5 ± 2.8	15.2 ± 2.6	0.620	0.327	0.642
Ankle-brachial index (mean)	1.14 ± 0.07	1.12 ± 0.07	1.12 ± 0.08	0.565	0.344	0.662
**VerifyNow P2Y12 assay**						
PRU	138 ± 77	130 ± 64	214 ± 50	0.178	<0.001	<0.001
BASE	218 ± 39	208 ± 35	206 ± 36	0.102	0.133	0.639
**Thromboelastography**						
R, minutes	5.8 ± 1.3	5.5 ± 1.2	5.1 ± 1.1	0.379	0.017	0.037
K, minutes	1.9 ± 0.8	1.9 ± 0.7	1.6 ± 0.5	0.727	0.124	0.157
Angle, degree	61.4 ± 8.5	62.9 ± 9.0	65.2 ± 8.1	0.441	0.145	0.346
MA_thrombin_, mm	60.3 ± 5.9	59.9 ± 5.2	61.0 ± 5.2	0.748	0.603	0.388
LY30, %	1.2 ± 1.7	1.1 ± 1.7	1.4 ± 2.4	0.684	0.722	0.490

AI, augmentation index; ASP, aspirin; CLPD, clopidogrel; DAPT, dual antiplatelet therapy; DBP, diastolic blood pressure; PAT, peripheral arterial tonometry; PP, pulse pressure; PRU, P212 reaction unit; PWV, pulse wave velocity; RHI, reactive hyperemia index; SBP, systolic blood pressure.

**Table 4 jcm-10-02720-t004:** Determinants of normal endothelial function (RHI ≥ 2.1).

	Univariate Analysis			Multivariate Analysis		
	OR	95% CI	*p*	OR	95% CI	*p*
**Arterial stiffness indices**						
Central SBP ≥ 121 mmHg	4.687	1.240–17.715	0.016	3.780	0.605–23.634	0.155
Central PP ≥ 63 mmHg	3.606	1.367–9.513	0.010	0.939	0.225–3.924	0.931
PWV ≥ 14.5 m/sec	5.538	1.676–18.298	0.005	3.834	0.887–16.581	0.072
**VerifyNow test**						
PRU ≤ 132	3.555	1.307–9.621	0.013	4.028	1.087–14.925	0.037
**Thromboelastography**						
Angle ≤ 68 degree	9.414	2.010–44.087	0.004	7.420	1.445–38.099	0.016
LY30 ≥ 1.5%	3.417	1.208–9.662	0.021	2.790	0.755–10.312	0.124

Continuous variables were converted into optimal cut-off values by ROC curve analysis. CI, confidential interval; OR, odds ratio; PP, pulse pressure; PRU, P2Y12 reaction unit; PWV, pulse wave velocity; SBP, systolic blood pressure.

## Data Availability

The data presented in this study are available on request from the corresponding author. The data are not publicly available due to regulatory restrictions.

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
