# Peer review of "Effects of Monotherapy with Clopidogrel vs. Aspirin on Vascular Function and Hemostatic Measurements in Patients with Coronary Artery Disease: The Prospective, Crossover I-LOVE-MONO Trial"

_jcm, 2021, doi:10.3390/jcm10122720_

Round 1

Reviewer 1 Report

In the manuscript “Effects of monotherapy with clopidogrel vs. aspirin on vascular function and hemostatic measurements in patients with coronary artery disease: The prospective, crossover I-LOVE-MONO trial”, the author aimed to investigate the effect of clopidogrel- vs. aspirin-monotherapy on vascular function and hemostatic measurement. The topic of the manuscript is certainly interesting and captures one of the issues of the moment on secondary cardiovascular prevention in high-risk patient.

The aim of the manuscript is actual, however, there are some points that need further clarification:

  1. The authors defined patients as “The enrolled DES-treated patients…”. Please specify that the patients underwent percutaneous coronary intervention with angioplasty and stent implantation.

  1. No mention was made of the index event type. Patients were revascularized with DES implantation, but were they stable patients or patients with acute coronary syndrome? This obviously makes a difference on the type of DAPT, duration of DAPT, and alterations in endothelial function and thrombotic burden. Please, provide this information.

  1. Please edit Figure 1 to make it more readable and pleasing to readers.

  1. The authors state that long-term use of potent P2Y12 inhibitors increases the risk of bleeding in the Asian population. Please briefly add to the discussion, early real-life results on the use of ticagrelor 60 mg, as this results in safety in the observed population (ref. Low-dose Ticagrelor in Patients With High Ischemic Risk and Prior Myocardial Infarction: a Multicenter Prospective Real-World Observational Study. J Cardiovasc Pharmacol. 2020 Jun 17.doi: 10.1097/FJC.0000000000000856).

  1. Please, briefly discuss the recent findings from real-world registry that showed that in high bleeding risk patients, no difference in NACEs was observed between patients on clopidogrel versus ticagrelor(Clopidogrel versus ticagrelor in high‑bleeding risk patients presenting with acute coronary syndromes: insights from the multicenter START‑ANTIPLATELET registry, Intern Emerg Med (2020) https://doi.org/10.1007/s11739-020-02404-1).

Author Response

Response to Comments from Reviewer 1

In this manuscript, the author aimed to investigate the effect of clopidogrel- vs. aspirin-monotherapy on vascular function and hemostatic measurement. The topic of the manuscript is certainly interesting and captures one of the issues of the moment on secondary cardiovascular prevention in high-risk patient. The aim of the manuscript is actual, however, there are some points that need further clarification:

We thank for the reviewer’s comments, recognizing the strengths and weakness of our study. The points of concern are addressed in detail below, as presented in a significantly revised version of the manuscript.

1. The authors defined patients as “The enrolled DES-treated patients…”. Please specify that the patients underwent percutaneous coronary intervention with angioplasty and stent implantation.

>>> Response: We revised this sentence according to the reviewer’s comment (line 88).

2. No mention was made of the index event type. Patients were revascularized with DES implantation, but were they stable patients or patients with acute coronary syndrome? This obviously makes a difference on the type of DAPT, duration of DAPT, and alterations in endothelial function and thrombotic burden. Please, provide this information.

>>> Response: Thanks for the reviewer’s comment. Table 2 suggested that 80.8% of patients were presented with MI. We revised Table 2 showing initial presentation type (page 9). Patients presented with stable angina were treated with DAPT less than 12 months.

3. Please edit Figure 1 to make it more readable and pleasing to readers.

>>> Response: We changed Figure 1. Hope it would be more readable to readers (page 7).

4. The authors state that long-term use of potent P2Y12 inhibitors increases the risk of bleeding in the Asian population. Please briefly add to the discussion, early real-life results on the use of ticagrelor 60 mg, as this results in safety in the observed population (ref. Low-dose Ticagrelor in Patients With High Ischemic Risk and Prior Myocardial Infarction: a Multicenter Prospective Real-World Observational Study. J Cardiovasc Pharmacol. 2020 Jun 17.doi: 10.1097/FJC.0000000000000856).

>>> Response: According to the reviewer’s comment, we added this finding and PEGASUS-TIMI 54 result in the Discussion section (page 12).

■ Addition of the manuscript, Discussion, page 12:

In the randomized PEGASUS-TIMI 54 (Prevention of Cardiovascular Events in Patients With Prior Heart Attack Using Ticagrelor Compared to Placebo on a Background of Aspirin – Thrombolysis in Myocardial Infarction 54) trial, the use of low-dose ticagrelor (60 mg twice a day) plus aspirin after 12 months from MI reduced the risk of ischemic events, at the price of limited increase on bleeding complications [20]. In addition, a recent Western real-world observational study provided that low-dose ticagrelor after 12 months from MI showed to be effective and safe, with no major bleeding occurring at follow-up [21].

 5. Please, briefly discuss the recent findings from real-world registry that showed that in high bleeding risk patients, no difference in NACEs was observed between patients on clopidogrel versus ticagrelor(Clopidogrel versus ticagrelor in highbleeding risk patients presenting with acute coronary syndromes: insights from the multicenter STARTANTIPLATELET registry, Intern Emerg Med https://doi.org/10.1007/s11739-020-02404-1).

>>> Response: Thanks for the reviewer’s comment. Actually, Korean AMI registry also showed no benefit in ischemic event during potent P2Y12 inhibitors vs. clopidogrel (PMID: 29534250). The START-ANTIPLATELET registry only suggested clinical outcome during 1 year following PCI (DAPT with clopidogrel vs. ticagrelor in HBR patients), whereas I-LOVE-MONO only focused on long-term treatment after standard DAPT maintenance (after 9-12 months). Hope we can discuss this issue regarding optimal DAPT regimen within 1 year post-PCI in HBR patients.

Reviewer 2 Report

Thank you for the opportunity to review this very well written manuscript of a very well conducted I-LOVE-MONO trial. Despite the limitations mentioned by authors in the limitation section about small sample size and short duration of intervention and lack of knowledge of long-term effect on vascular or hemostatic function of short and/or longer duration of antiplatelet therapy, the current study has potential for promising future and a novel as well as first finding of it's kind. Further trial with larger sample size and different duration of intervention, as well as "real" clinical outcomes assessment is needed for practice changing recommendations. Thank you.

Author Response

Thank you for the opportunity to review this very well written manuscript of a very well conducted I-LOVE-MONO trial. Despite the limitations mentioned by authors in the limitation section about small sample size, short duration of intervention and lack of knowledge of long-term effect on vascular or hemostatic function of short and/or longer duration of antiplatelet therapy, the current study has potential for promising future and a novel as well as first finding of it's kind. Further trial with larger sample size and different duration of intervention, as well as "real" clinical outcomes assessment is needed for practice changing recommendations.

We thank for the reviewer’s comments, recognizing the strengths and weakness of our study. Recently published HOST-EXAM trial suggested the better efficacy and safety result by clopidogrel- vs. aspirin-monotherapy, which would be a big asset to change recommendation in the near future.